# The Regulation of Rab GTPases by Phosphorylation

**DOI:** 10.3390/biom11091340

**Published:** 2021-09-10

**Authors:** Lejia Xu, Yuki Nagai, Yotaro Kajihara, Genta Ito, Taisuke Tomita

**Affiliations:** 1Laboratory of Neuropathology and Neuroscience, Graduate School of Pharmaceutical Sciences, The University of Tokyo, Tokyo 113-0033, Japan; xulejia@g.ecc.u-tokyo.ac.jp (L.X.); nagai-yuki@g.ecc.u-tokyo.ac.jp (Y.N.); youtarou-bad-0221@g.ecc.u-tokyo.ac.jp (Y.K.); 2Department of Biomolecular Chemistry, Faculty of Pharma-Science, Teikyo University, Tokyo 173-8605, Japan; 3Social Cooperation Program of Brain and Neurological Disorders, Graduate School of Pharmaceutical Sciences, The University of Tokyo, Tokyo 113-0033, Japan

**Keywords:** Rab, phosphorylation, membrane trafficking, LRRK1, LRRK2, TBK1

## Abstract

Rab proteins are small GTPases that act as molecular switches for intracellular vesicle trafficking. Although their function is mainly regulated by regulatory proteins such as GTPase-activating proteins and guanine nucleotide exchange factors, recent studies have shown that some Rab proteins are physiologically phosphorylated in the switch II region by Rab kinases. As the switch II region of Rab proteins undergoes a conformational change depending on the bound nucleotide, it plays an essential role in their function as a ‘switch’. Initially, the phosphorylation of Rab proteins in the switch II region was shown to inhibit the association with regulatory proteins. However, recent studies suggest that it also regulates the binding of Rab proteins to effector proteins, determining which pathways to regulate. These findings suggest that the regulation of the Rab function may be more dynamically regulated by phosphorylation than just through the association with regulatory proteins. In this review, we summarize the recent findings and discuss the physiological and pathological roles of Rab phosphorylation.

## 1. Introduction: The Potential Role of Phosphorylation in the Regulation of Rab

The Rab family is a member of the Ras small GTPase superfamily, which consists of more than 60 proteins in humans (reviewed in [1]). Rab family proteins share a common function in the regulation of intracellular vesicle transport. Similar to other small GTPases, Rab GTPases exist intracellularly as a GTP-bound or GDP-bound form and function as a molecular switch for membrane trafficking. Most Rab proteins are expressed ubiquitously throughout the body, but some Rab proteins are known to function in specific cells, in which they regulate the transport of specialized vesicles. For example, Rab38 in melanocytes regulates the transport of melanosomes, which specifically exist in melanocytes.

When a Rab protein is biosynthesized, it is first recognized by Rab escort proteins (REPs) in the cytosol and then recruited to the Rab geranylgeranyl transferase (RabGGTase) complex, by which it undergoes geranylgeranylation modification [2,3] (Figure 1A). Geranylgeranylation is a type of post-translational modification of proteins in which a geranylgeranyl group, a 20-carbon hydrophobic group, is transferred to the thiol group of a cysteine residue. The Rab protein geranylgeranylated at carboxyl-terminal cysteine residue(s) is inserted into the lipid bilayer by REPs [4].

Once inserted into the lipid bilayer, each Rab controls different vesicular trafficking pathways depending on its localization. The localization of a Rab protein is thought to be defined by its activator, a guanine nucleotide exchange factor (GEF). GEF activity accelerates the dissociation of GDP from the Rab protein, and this then induces the binding of cytoplasmically abundant GTP to the Rab protein (reviewed in [5]). The GTP-bound Rab protein initiates vesicular transport by binding to its effector proteins that selectively bind to its GTP-bound form (Figure 1A).

The destination of vesicular trafficking is defined by multiple factors. One factor is the combination of a soluble N-ethylmaleimide sensitive factor attachment protein receptor (SNARE) protein present on the vesicle membrane (v-SNARE) and multiple SNARE proteins on the target membrane (t-SNARE) that promotes the fusion of the membranes (reviewed in [6]). Another is the existence of a GTPase-activating protein (GAP) that increases the GTPase activity of the Rab protein, thereby switching off the transport. By binding to GAPs, the bound GTP is hydrolyzed to GDP (reviewed in [7]). The GDP-bound Rab is then extracted from the lipid bilayer into the cytosol by GDP-dissociation inhibitors (GDIs), which have a high affinity for GDP-bound geranylgeranylated Rab proteins. Thereafter, the Rab is recycled back to the donor membrane (reviewed in [8]) (Figure 1A).

Although it has been textbook knowledge that Rab is regulated by GEFs, GAPs and GDIs, as outlined above, recent studies have revealed that the Rab function is also regulated by phosphorylation. The phosphorylation of Rab proteins has been shown not only to inhibit their interaction with GEFs and GDIs, but also to let them acquire different effector proteins, suggesting the existence of alternative trafficking routes activated by Rab phosphorylation (Figure 1B,C). In this review, we summarize the recent discoveries on Rab phosphorylation and its relevance to biology and disease.

## 2. Rab Phosphorylation by LRRK2

### 2.1. Discovery of Rab Phosphorylation by LRRK2

Leucine-rich repeat kinase 2 (LRRK2) is one of the responsible gene products for familial Parkinson’s disease (PD) [9,10]. PD is the second most common neurodegenerative disease, pathologically characterized by the appearance of a cytoplasmic inclusion body, called Lewy bodies, throughout the degenerating regions and a selective loss of dopaminergic neurons in substantia nigra in the midbrain [11,12,13,14]. The genomic locus encoding LRRK2 has also been shown to be associated with an increased risk of developing a non-hereditary form of PD [15,16]. These findings indicate that LRRK2 plays a major role in the pathogenesis of PD.

LRRK2 is a large protein kinase consisting of 2527 amino acids, harboring a GTP-binding domain called a Ras-of-complex proteins (ROC) domain followed by a carboxyl-terminal of the ROC (COR) domain, an auxiliary domain of the ROC domain and a protein kinase domain [17] (Figure 2). Pathogenic mutations linked with familial PD (e.g., N1437H, R1441C/G/H/S, Y1699C, G2019S, I2020T) have been found in these domains. As the G2019S mutation, the most frequent pathogenic mutation, substantially increases the LRRK2 kinase activity [18], it has been proposed that an abnormal increase in the phosphorylation of LRRK2 substrates causes neurodegeneration in PD.

In 2016, Steger and colleagues discovered that several Rab proteins are phosphorylated by LRRK2 under physiological conditions [19]. They identified Rab10 and Rab12 as substrates of LRRK2 by a combination of phosphoproteomics, genetics and pharmacology in mouse embryonic fibroblasts (MEFs), and then provided further evidence that Rab8 and Rab12 are endogenously phosphorylated by LRRK2 in mouse brains. The overexpression of these Rab proteins with any familial mutant forms of LRRK2 showed a marked increase in their phosphorylation. The excessive phosphorylation of Rab10 at Thr73 by R1441G and G2019S LRRK2 was later confirmed by the same group at endogenous levels in the lung as well as in MEFs using knock-in mouse models [20].

Steger and colleagues further showed that LRRK2 phosphorylates fourteen Rab proteins, namely Rab3A/B/C/D, Rab5A/B/C, Rab8A/B, Rab10, Rab12, Rab29, Rab35 and Rab43, upon co-overexpression in HEK293 cells [21] (Figure 2). Among these fourteen Rab proteins, the phosphorylation of Rab3A/B/C/D, Rab8A/B, Rab10, Rab35 and Rab43 was confirmed at endogenous levels in MEFs harboring the R1441C LRRK2 knock-in mutation.

In a paper published in 2019 by Berndsen and colleagues, the endogenous phosphorylation of Rab10 and Rab12 in R1441C LRRK2 knock-in MEFs was shown by immunoblotting using phospho-specific antibodies [22]. In this report, they also identified Protein phosphatase 1H (PPM1H) as a phosphatase responsible for the endogenous dephosphorylation of Rab10. This is an important initial finding, as enzymatic reversibility is a critical feature of regulatory modifications. In the future, it will be important to investigate the dephosphorylation process of Rab in more depth in order to obtain a complete picture of the regulation of Rab proteins by phosphorylation.

A similar result was reported by other groups. Jeong and colleagues showed the phosphorylation of a panel of Rab proteins by recombinant wild-type (WT) LRRK2 in an in vitro assay using isotope-labelled ATP [23]. They found that several Rab proteins, namely Rab1A/B, Rab3C, Rab8A/B, Rab9B, Rab10, Rab23 and Rab35 were phosphorylated by LRRK2. The phosphorylation of Rab29 (also known as Rab7L1) upon co-overexpression with LRRK2 in HEK293 cells was also reported by Fujimoto and colleagues [24], which is discussed in a section below.

Based on these results, it is now widely accepted that LRRK2 is a Rab kinase, and its physiological and pathological relevance has attracted much attention.

### 2.2. Rab Phosphorylation by LRRK2 in Human Samples

Since the discovery of Rab phosphorylation by LRRK2, researchers have been focusing on detecting Rab phosphorylation in human body fluid samples as a potential method for diagnosis. Thirstrup and colleagues employed a combined proteomics and phosphoproteomics analysis on human peripheral mononuclear cells (PBMCs) and identified the LRRK2-dependent phosphorylation of Rab10 and Rab12 at Thr73 and Ser106, respectively [25]. As PBMCs are quite heterogeneous, consisting of lymphocytes and monocytes, and the expression levels of LRRK2 as well as the substrate Rab proteins are different among these types of cells, the levels of the LRRK2-dependent phosphorylation of Rab proteins in PBMCs should depend not only on the LRRK2 activity but also on the proportion of cells expressing LRRK2 and Rab at high levels.

In a more recent report by Fan and colleagues, the authors compared various types of blood cells and found, by immunoblotting, that neutrophils are superior to PBMCs in the detection of Rab10 phosphorylation [26]. As neutrophils are abundantly present in human peripheral blood, the volume of blood required for the detection of Rab10 phosphorylation can be reduced, which increases the feasibility of sample collection at local clinics. Importantly, the authors presented a slight but significant increase in the Rab10 phosphorylation on immunoblots in idiopathic PD patients compared to healthy control individuals [26]. Atashrazm and colleagues also characterized Rab10 phosphorylation in the PBMCs and neutrophils isolated from human peripheral blood [27]. They also compared Rab10 phosphorylation between healthy controls and PD patients, but there was no significant difference.

Another report by Karayel and colleagues failed to find a difference in the levels of Rab10 phosphorylation in neutrophil samples between healthy controls and idiopathic PD patients, either using immunoblotting or using mass spectrometry [28]. However, they found a significant increase in Rab10 phosphorylation in G2019S LRRK2 as well as D620N VPS35 mutation carriers. A recent follow-up study from Fan and colleagues also showed no difference between the controls and the idiopathic PD patients, although a significant increase could be detected in patients harboring the R1441G LRRK2 mutation [29]. In a recent report by Wang and colleagues, they established a quantitative and high-throughput assay to measure the levels of Rab10 phosphorylation in human PBMCs [30]. However, the levels of Rab10 phosphorylation in controls and PD patients were not significantly different, although they detected a significant increase in the Rab10 phosphorylation in G2019S carriers compared with non-carriers, regardless of the disease status.

Given that neurodegeneration occurs in PD, it is important to investigate whether Rab10 phosphorylation can be detected in the brain and whether there is a difference between the controls and the PD patients. Di Maio and colleagues examined the Rab10 phosphorylation in postmortem brains using immunohistochemistry [31]. They detected a substantial increase in the intensity of the phospho-Rab10 staining in the brains of idiopathic PD patients. On the other hand, Fan and colleagues examined the Rab10 phosphorylation in postmortem brains using immunoblotting and mass spectrometry but failed to show any difference between the controls and the PD patients [29].

The reason for this discrepancy between reports is not clear, but it may be due to the varying levels of Rab10 phosphorylation among individuals. Therefore, further studies with a larger number of subjects and different cohorts are required to clarify the usefulness of Rab10 phosphorylation in diagnosing PD using blood samples, as well as to examine the change in Rab10 phosphorylation in PD patient brains.

### 2.3. Functional Significance of Rab Phosphorylation by LRRK2

Rab proteins are phosphorylated by LRRK2 via the highly conserved switch II region, which regulates the hydrolysis of GTP and coordinates the binding to various regulatory proteins [8]. Therefore, phosphorylation by LRRK2 is considered to have an inhibitory effect on the interaction between Rabs and their regulatory proteins, such as GEFs and GAPs, as well as effector proteins [19]. However, there is also a case in which phosphorylated Rab proteins specifically bind to certain regulatory proteins to which they do not bind in their non-phosphorylated forms. To date, eight proteins, namely EH domain-binding protein 1 (EHBP1), EHBP1-like protein 1 (EHBP1L1), Rab-interacting lysosomal protein (RILP), RILP-like protein 1 (RILPL1), RILP-like protein 2 (RILPL2), c-Jun-amino-terminal kinase (JNK)-interacting protein 3 (JIP3), JNK-interacting protein 4 (JIP4) and Folliculin (FLCN)-FLCN-interacting protein 1 (FNIP1), are proposed as effector proteins that specifically bind to phosphorylated Rab proteins (Figure 3A) [21,32,33,34,35]. EHBP, EHBP1, RILPL1, RILPL2, JIP3 and JIP4 are the effector proteins of phosphorylated Rab8, Rab10 and Rab12, whereas RILP and FLCN-FNIP1 are those of phosphorylated Rab7A. For example, the interaction between Rab8A and RILPL2 was observed only when Rab8A was phosphorylated by LRRK2 [21]. From these observations, it was hypothesized that switching effector proteins, depending on the phosphorylation by LRRK2, might activate a currently unknown alternative trafficking route regulated by the Rab protein (Figure 1C). Further investigations are required to clarify whether the phosphorylation of Rab proteins plays a major role in determining effector proteins and transport pathways.

The phosphorylation of Rab8 and Rab10 by LRRK2 is reported to play an important role in lysosomal homeostasis. The treatment of cells with lysosomotropic agents (e.g., chloroquine and L-Leucyl-L-Leucine methyl ester (LLOMe)) results in lysosomal overload, in which lysosomes become enlarged. In mouse macrophagic Raw264.7 cells, in response to the overload, lysosomal contents are actively released to the exterior of the cells. Intriguingly, Eguchi and colleagues have shown that the phosphorylation of Rab8 and Rab10 by LRRK2 is required for this release, as the treatment of cells with LRRK2 inhibitors or knockdown of Rab8 or Rab10 caused an enlargement of lysosomes and a concomitant decrease in the amount of extracellularly released Cathepsin D, a lysosomal enzyme [32,36,37]. These observations indicate that the phosphorylation of Rab8 and Rab10 plays an essential role in the stress response against lysosomal overload in macrophages (Figure 3B).

In macrophages, the activation of LRRK2 is also observed upon endomembrane damage caused by an infection of pathogens or lysosomotropic drug treatment [38]. Endomembrane damage triggers the LRRK2-dependent phosphorylation of Rab8A, followed by the recruitment of Charged multivesicular body protein 4B (CHMP4B), a component of the endosomal sorting complex required for transport III, or ESCRT-III, to LRRK2/Rab8A-positive damaged lysosomes. It is assumed that ESCRT-III helps to repair damaged lysosomes, which prevents lysosomes from being degraded by lysophagy (Figure 3B). An experiment using mouse primary astrocytes gave a similar result where LRRK2 is recruited to lysosomes as a response to lysosomal membrane rupture [39]. In this experiment, the lysosomal membrane rupture induced by treatment with a lysosomotropic agent resulted in the LRRK2-dependent phosphorylation of Rab10 and Rab35 on the damaged lysosomal membranes, followed by the recruitment of JIP4 (also known as SPAG9) via the interaction with these Rab proteins. JIP4 promotes the formation of lysosome-associated membrane glycoprotein 1 (LAMP1)-negative tubular structures extending from the damaged lysosomes along microtubules. The tubules eventually generate vesicles that can interact with other lysosomes. This phenomenon was named LYsosomal Tubulation/sorting driven by LRRK2, or LYTL, and considered a potentially novel sorting process of the vesicles budded from lysosomes (Figure 3B).

There is also a study reporting that LRRK2 phosphorylates Rab10 on early immature macropinosomes in macrophages [40]. Rab10 on macropinosomes normally binds to the effector protein EHBP1L1 and thereby promotes the fast recycling of macropinosomes back to the plasma membrane, and this happens in an LRRK2-independent manner. However, the LRRK2-mediated phosphorylation of Rab10 inhibits its binding to EHBP1L1, thereby reducing the recycling of macropinosomes. Instead of EHBP1L1, phosphorylated Rab10 recruits the effector protein RILPL2. Additionally, the Rab10-positive macropinosomes were positive for a chemokine receptor, CCR5. Given these observations, the authors proposed that, in macrophages, switching the effector from EHBP1L1 to RILPL2 depending on Rab10 phosphorylation slows the fast recycling of CCR5, and through a yet-to-be-determined mechanism that may involve Akt phosphorylation, enables the cells to amplify the chemokine signal, thereby facilitating chemotaxis to promptly respond to chemokines (Figure 3C).

### 2.4. Potential Role of LRRK2-Rab in the Formation of Primary Cilia

Several studies have reported the potential role that the LRRK2-Rab pathway plays in blocking the formation of primary cilia. A prerequisite for primary cilia formation is the dissociation of centrosomal protein of 110 kDa (CP110) from the mother centriole, which becomes the basal body in ciliogenesis. Tau-tubulin kinase 2 (TTBK2) is a kinase required for this dissociation, and it is recruited to the pericentriolar region through the interaction with centrosomal protein 164 (CEP164), a centriole protein. It was shown that this recruitment of TTBK2 to the pericentriolar region is inhibited by the hyperphosphorylation of Rab10 [41] (Figure 3D). In addition, Dhekne and colleagues have found that the subcellular localization of Rab10 is changed to the pericentriolar region upon phosphorylation by LRRK2 in a manner dependent on the presence of RILPL1 [42]. RILPL1 has been reported to associate with the mother centriole [43], and it has also been shown that RILPL1 binds to phosphorylated Rab10 [21]. Collectively, these results suggest that the Rab10 phosphorylated by LRRK2 is recruited to the pericentriolar region together with RILPL1 (and potentially RILPL2). This recruitment may lead to the inhibition of ciliogenesis, possibly by inhibiting the dissociation of CP110. Sobu et al. suggested that this inhibition may occur through a yet-to-be-determined mechanism that involves phosphatidylinositol phosphates, which were proposed to regulate ciliogenesis through regulating the interaction of TTBK2 with CEP164. Furthermore, LRRK2-phosphorylated Rab10 causes the aberrant translocation of Myosin Va to mother centrioles, which might interfere with the function of the Myosin Va required for ciliogenesis [44]. In these ways, LRRK2 and Rab phosphorylation may interfere with primary cilia formation.

This link between LRRK2 and ciliogenesis may also affect the sonic hedgehog (Shh) signaling. The Shh signaling requires the formation of primary cilia, to which the downstream transcription factors, such as Gli1 and Gli2, are localized [45,46]. LRRK2 R1441G knock-in cells, as well as iPS cells derived from PD patients carrying the LRRK2 G2019S mutation, exhibited defects in ciliation and a decrease in the Shh signaling, indicating that the abnormal activation of LRRK2 blocks the Shh signaling in cells [42]. In addition, they found that LRRK2 R1441C knock-in mice showed ciliation defects in certain brain regions including cholinergic neurons in the striatum. Striatal cholinergic neurons receive the Shh signals from nigrostriatal dopaminergic neurons [47], and in return, send neuroprotective signals back to dopaminergic neurons in the substantia nigra [48]. Therefore, the loss of dopaminergic neurons in LRRK2-mediated PD may be a result of poor ciliogenesis in striatal cholinergic neurons.

### 2.5. Structural Evidence of the Effects of Rab Phosphorylation on the Effector Binding

The effects of Rab phosphorylation on its interaction with its effector have also been investigated in the context of structural biology. In 2020, Waschbüsch and colleagues reported the structure of phosphorylated Rab8A (pRab8A) in complex with the RH2 domain of RILPL2 using X-ray crystallography [33]. RILPL2 and pRab8A formed a heterotetramer within the crystal. Within the complex, RILPL2 formed an α-helical dimer, and this dimer oriented over two pRab8A molecules in a way that bridged over them. The amino-termini of this dimer formed a unique X-shaped structure called the X-cap. This X-cap enabled the interaction between the Arg132 of RILPL2 and the phosphorylated Thr72 (pThr72) of Rab8A. In addition, Arg130 also interacted with pThr72, further enhancing the stability of the binding between pRab8A and RILPL2 [49]. These results provide structural biological support for the effect of Rab phosphorylation on its binding to effector proteins.

### 2.6. Proposed Roles of LRRK2-Rab in α-Synuclein Propagation

In PD, the aggregation of α-synuclein into inclusion bodies called Lewy bodies is a prominent pathological feature [13,14]. It has been hypothesized that α-synuclein possesses prion-like properties, in which it can propagate through the brain, triggering α-synuclein in the surrounding brain areas to aggregate [50]. It has been shown that α-synuclein pathology initiates in the brainstem and olfactory bulb before motor symptoms arise, and progressively spreads to connected brain areas such as the substantia nigra pars compacta (SNpc), eventually reaching the neocortex [51]. The propagation of α-synuclein has been experimentally shown in animals injected with pre-formed α-synuclein fibrils into the brain (reviewed in [52]).

A number of studies have looked into the effects of LRRK2 mutations on the accumulation of α-synuclein. In these reports, dopaminergic neurons differentiated from induced pluripotent stem cells (iPSCs) prepared from PD patients harboring the G2019S LRRK2 mutation consistently showed increased amounts of cytoplasmic α-synuclein, presumably due to posttranslational alterations [53,54,55,56]. Among these reports, Sánchez-Danés and colleagues showed an accumulation of autophagic vacuoles and lipid droplets in differentiated dopaminergic neurons harboring the G2019S LRRK2 mutation, suggesting the impairment of autophagosome clearance in these neurons. The increase in the levels of α-synuclein accumulation by G2019S LRRK2 was recapitulated in the brains of transgenic rats harboring bacterial artificial chromosome (BAC) encoding G2019S LRRK2, which were administered with pre-formed α-synuclein fibrils [57]. This result was consistent with the previous report in rats in which α-synuclein accumulation in the substantia nigra by the AAV-mediated overexpression of α-synuclein was attenuated in LRRK2-deficient rats [58].

Numerous studies have also looked into the roles that LRRK2 and Rab may play in α-synuclein propagation. However, there has been conflicting evidence, and it remains controversial whether or not LRRK2 is involved in α-synuclein pathology.

In 2018, Henderson and colleagues reported that LRRK2 activity only has minor effects on α-synuclein pathology [59]. Primary neurons cultured from BAC transgenic mice expressing the pathological mutant G2019S LRRK2 did not exhibit any dramatic increase in the pathological α-synuclein burden. Moreover, all of the three LRRK2 inhibitors tested failed to reduce the α-synuclein pathology induced by the treatment of neurons with pre-formed α-synuclein fibrils. In addition to hippocampal neurons, neither G2019S LRRK2 expression nor LRRK2 inhibition altered α-synuclein pathology in midbrain dopaminergic neurons as well. The authors concluded that these data suggest that LRRK2 activity has no more than minor effects on α-synuclein pathology in primary neurons.

In a paper published in 2019, the same group further reported that in non-transgenic mice, LRRK2 inhibition does not lead to increased protection from α-synuclein pathology or neuron death [60]. They made use of a mouse model of PD, created by an injection of a small amount of pathogenic α-synuclein into the striatum. This model recapitulates the spread of pathological α-synuclein inclusions and the consequent degeneration of dopaminergic neurons in the substantia nigra. Importantly, LRRK2 inhibition by the potent LRRK2 inhibitor MLi-2 did not alter motor phenotypes, pathological α-synuclein accumulation, nor neuron loss. The lack of change in α-synuclein accumulation following MLi-2 treatment was observed not only in the substantia nigra but also in other regions of the brain, such as the ventral tegmental area, hippocampus, posteromedial cortical amygdaloid nucleus and the visual cortex. This led the authors to suggest that LRRK2 does not affect α-synuclein pathogenesis in this mouse model of PD, and that further studies are necessary to evaluate the benefit of LRRK2 inhibition in idiopathic PD.

On the other hand, Bieri and colleagues published a paper in 2019 reporting that LRRK2 does modify α-synuclein pathology and spread in mouse models and human neurons [61]. They performed a targeted genetic screen of risk genes related to PD and parkinsonism to identify modifiers of α-synuclein aggregation. They showed that the knockdown (KD) of Lrrk2 resulted in a significant decrease in α-synuclein aggregation. This effect was further confirmed with Lrrk2 KD using *Lrrk2*-targeting guide RNAs in primary neurons cultured from Cas9 transgenic mice. Conversely, in primary neurons from transgenic mice expressing G2019S LRRK2, an increase in α-synuclein aggregation was observed. Furthermore, LRRK2 G2019S transgenic mice exhibited an acceleration of α-synuclein aggregation and the degeneration of dopaminergic neurons in the SNpc increased degeneration-associated neuroinflammation and behavioral deficits. This was further validated in a human context by establishing a human α-synuclein transmission model using iPSC-derived neurons (iNs). In the iNs derived from the PD subjects, the G2019S mutation led to the enhanced aggregation of α-synuclein, while the loss of LRRK2 decreased it. Altogether, the authors suggested a strong link between the LRRK2 activity and α-synuclein propagation.

In 2018, Bae and colleagues also reported that LRRK2 plays a role in α-synuclein propagation, proposing a more detailed mechanism in which one of the substrates of LRRK2, Rab35, participates in α-synuclein propagation [62]. They showed that in nematode and rodent models of PD, deficiency of the LRRK2 gene led to reduced α-synuclein propagation. They also demonstrated that the G2019S mutation increases the efficiency of α-synuclein propagation, using a dual-cell bimolecular fluorescence complementation system in which α-synuclein is secreted from one cell line and internalized by another. Additionally, α-synuclein partially colocalized with overexpressed Rab35, a substrate of LRRK2. Rab35 mutant *Caenorhabditis elegans* exhibited reduced levels of α-synuclein aggregation compared to the wild-type, and this could be rescued with the ectopic expression of Rab35, but not with the T72A mutant of Rab35, which cannot be phosphorylated by LRRK2. Furthermore, the reduced α-synuclein propagation phenotype in *lrk-1* mutant *C. elegans* could be overridden with the expression of the constitutively activated Q67L mutant of RAB35, accompanied with worsened degenerative phenotypes including nerve degeneration and life span. Lastly, using an α-synuclein transgenic mouse model, the authors showed that the administration of an LRRK2 kinase inhibitor led to a decrease in α-synuclein aggregation, accompanied by an increasing interaction of α-synuclein with the lysosomal enzyme Cathepsin D, suggesting an enhanced association with the lysosomal degradation pathway. This prompted the authors to suggest that the LRRK2-mediated RAB35 phosphorylation positively regulates α-synuclein propagation.

Collectively, although the above studies examining the expression levels of α-synuclein in iPSC-derived dopaminergic neurons appear to conclude that G2019S LRRK2 exacerbates α-synuclein accumulation, it remains controversial whether LRRK2 plays a role in the propagation of α-synuclein pathology, especially in an in vivo setting. The reason for the discrepancies is unclear at the moment, but it may be due to multiple factors including differences in the methods for the preparation of α-synuclein seeds (using 100 mM NaCl at pH 7.0 in [59,60] vs. using 150 mM KCl at pH 7.5 in [61]) as well as differences in the mouse models used (WT mice + pre-formed α-synuclein fibrils +/− LRRK2 inhibitor (MLi-2) in [60], WT or BAC G2019S transgenic mice + pre-formed α-synuclein fibrils in [61], α-synuclein transgenic mice +/− LRRK2 inhibitor (HG-10-102-01) in [62]). Alternatively, it would be interesting to hypothesize that G2019S LRRK2 but not WT LRRK2 gains a toxic function, thereby facilitating the propagation of α-synuclein pathology. In any case, it would be important to try to further narrow down the factor(s) that dictate the degree of involvement of LRRK2 in the propagation of α-synuclein pathology.

### 2.7. Involvement of Rab29 in the LRRK2-Rab Pathway

Rab29 (also known as Rab7L1) has been implicated in PD pathogenesis as several genome-wide association studies on sporadic PD found that a genomic locus containing Rab29 is associated with an increased risk of developing PD [15]. Intriguingly, besides the evidence in human genetics, several studies have pointed to a relationship between Rab29 and LRRK2.

Firstly, Rab29 is a physiological substrate of LRRK2. LRRK2 phosphorylates Rab29 at Thr71 or Ser72, or both, in the switch II region [21,24,63]. The Rab29 overexpressed in cultured cells is localized to the trans-Golgi network (TGN) and is involved in the maintenance of its morphology [64,65,66,67]. Fujimoto and colleagues showed that the overexpression of non-phosphorylatable S72A Rab29 caused the condensation of TGN in a similar manner to that by WT Rab29, whereas that of phosphomimetic S72E Rab29 did not, suggesting that the phosphorylation of Ser72 has an inhibitory role on the Golgi condensation [24].

Secondly, the overexpression of Rab29 in cultured cells activates LRRK2. When Rab29 was co-overexpressed in HEK293 cells with WT LRRK2, the autophosphorylation of LRRK2 at Ser1292 as well as Rab10 phosphorylation was dramatically increased [53]. This activation of LRRK2 by Rab29 was also observed when Rab29 was engineered to localize on mitochondrial membranes, indicating that the membrane identity is not an essential factor for the activation of LRRK2 [68]. In this experiment, LRRK2 was recruited to the mitochondrial membranes, suggesting that Rab29 recruits LRRK2 to membranes and promotes its substrate phosphorylation on the membrane.

The dependence of LRRK2 activity on Rab29 was also supported by a paper by Eguchi and colleagues [32]. As mentioned in a previous section, LRRK2 is recruited to overloaded lysosomes and facilitates the release of their contents in a manner dependent on Rab8 and Rab10. In this experimental setting, the knockdown of Rab29 downregulated the translocation of LRRK2 to enlarged lysosomes, promoted lysosomal enlargement and reduced the amount of Cathepsin D released into the culture media.

However, the mechanism of how Rab29 activates LRRK2 remains unclear. Several reports have shown that the knockout (KO) of *Rab29* in cells or mice has no impact on the phosphorylation of LRRK2 substrates. In a paper by Kalogeropulou and colleagues, they generated *Rab29* KO mice and found no effect on the phosphorylation of LRRK2 substrates in vivo [69]. Moreover, they also generated Rab29 transgenic mice that ubiquitously overexpress murine Rab29 under a phosphoglycerate kinase 1 promoter and found no effect on the phosphorylation of LRRK2 substrates. Kluss and colleagues also reported that the knockdown of Rab29 in HEK293FT cells overexpressing LRRK2 did not change the phosphorylation of Rab10 and Rab12 and proposed that the overexpression of Rab29 in cultured cells might cause membrane damage, resulting in the activation of LRRK2 and downstream pathways in a similar manner to what was observed in macrophages treated with lysosomotropic agents [70].

Collectively, it requires further investigation to elucidate the connection between Rab29 and LRRK2, especially in the context of PD pathogenesis.

### 2.8. Crosstalk with PINK1

Phosphatase and tensin homolog deleted from chromosome 10 (PTEN)-induced kinase 1 (PINK1) is a responsible gene for a type of autosomal recessive juvenile parkinsonism (AR-JP; PARK6). The loss-of-function mutations in PINK1 have been found in PARK6 patients. It has been well documented that PINK1 functions as a sensor of mitochondrial depolarization and facilitates the degradation of damaged mitochondria by recruiting and phosphorylating Parkin, another gene responsible for AR-JP (PARK2), as well as ubiquitin, leading to the degradation of the damaged mitochondria via autophagy (i.e., mitophagy) (reviewed in [71]). Therefore, loss of PINK1 function is thought to result in the accumulation of damaged mitochondria, which generates reactive oxygen species, resulting in neuronal death.

Lai and colleagues found that the phosphorylation of Rab1B, 8A, 8B and 13 is regulated by PINK1 [72]. These Rab proteins were not directly phosphorylated by PINK1 in an in vitro experiment, and the kinase responsible for the phosphorylation is currently unknown. Rab8A is phosphorylated by the PINK1-dependent kinase at Ser111, which is different from Thr72, the site phosphorylated by LRRK2. As the GEF-dependent dissociation of GDP from Rab8A was decreased by a phosphomimetic mutation at Ser111 (i.e., S111E), it is hypothesized that the PINK1-dependent phosphorylation of Rab8A has an inhibitory effect on the Rab8A function.

As Rab8A Ser111 is within the RabSF3 motif in close proximity to the switch II region containing Thr72, phosphorylation on either site may influence the efficiency of the other phosphorylation. Indeed, Rab8A harboring phospho-Ser111 seemed to be less efficiently phosphorylated by LRRK2 at Thr72 than WT Rab8A in vitro as well as in cultured cells [73]. These results suggested the potential crosstalk of the PINK1-dependent mitochondrial quality control and LRRK2 signaling, although its significance in the pathogenesis of PD requires further investigation.

## 3. Phosphorylation of Rab7A by LRRK1 and TBK1

### 3.1. Rab7A Phosphorylation by LRRK1

Leucine-rich repeat kinase 1 (LRRK1) is a paralog of LRRK2. These proteins share a similar domain architecture including ankyrin repeats, leucine-rich repeats, ROC, COR, a protein kinase domain and carboxyl-terminal WD40 repeats (Figure 2A,B). LRRK1 is smaller than LRRK2, as LRRK1 lacks the amino-terminal armadillo repeats. A low-resolution cryo-electron microscopic analysis on purified LRRK1 and LRRK2 has suggested that they form homodimers with a similar shape [74]. Both proteins bind GTP with their ROC domain in vitro, which is essential for their kinase activity [75,76,77].

It has been reported that LRRK1 regulates the mitotic spindle orientation by phosphorylating cyclin-dependent kinase 5 regulatory subunit-associated protein 2 (CDKRAP2), a centrosomal protein that plays a role in the maturation of centrosomes. The interaction of CDK5RAP2 with γ-tubulin was promoted upon phosphorylation by LRRK1, leading to an increase in the microtubule (MT) formation activity. When LRRK1 was knocked down, astral MTs were markedly decreased, and the MT nucleation activity of the centrosomes was downregulated, eventually leading to a loss of spindle orientation control [78].

LRRK1 was also shown to phosphorylate cytoplasmic linker protein 170 (CLIP-170, also known as CLIP1), a protein involved in dynein-mediated transport, thereby playing a role in EGFR trafficking [79]. CLIP-170 is one of the plus-end tracking proteins that link endosomes to MTs, and it directly binds to and recruits p150^Glued^/dynactin-1 to the MT plus ends. Additionally, activated EGFR is known to migrate on early endosomes along MTs as the endosomes undergo maturation. It was reported that the LRRK1-dependent phosphorylation of CLIP-170 enhances the interaction with p150^Glued^/dynactin-1, thereby facilitating the transport of EGFR [80].

Importantly, Hanafusa and colleagues recently provided evidence that shows that GTP-bound active-state Rab7A is also phosphorylated by LRRK1 at the endosomal membrane [35]. Rab7A is known to be localized to early endosomes and interact with the effector RILP, which recruits the dynein–dynactin motor complex, and facilitates the minus-end transport of endosomes. The phosphorylation of Rab7 by LRRK1 enhances the interaction of Rab7 with RILP, thereby promoting the centripetal trafficking of endosomes and its cargo such as EGFR (Figure 4A).

Malik and colleagues recently reported that LRRK1 and LRRK2 have distinct substrate spectra [77]. LRRK1 and LRRK2 are classified into the receptor-interacting protein (RIP) kinase family based on their structure [81]. RIP kinase family members function as sensors of intracellular and extracellular stresses. Given the similarity in structure and function among RIP kinases, it would be interesting to investigate whether RIP kinases other than LRRK1 and LRRK2 also phosphorylate Rab proteins.

Malik and colleagues also found that the phosphorylation of Ser72 in Rab7A was most significantly affected when LRRK1 was knocked out in MEFs, which suggests that LRRK1 is the major kinase for the phosphorylation of Ser72 in Rab7A, at least in MEFs. Of note, the authors showed that the phosphorylation of Rab7A did not change its interaction with RILP. Therefore, whether the effector protein of phosphorylated Rab7A plays a role in EGFR signaling remains controversial. Meanwhile, the authors also showed that stimulation of MEFs with phorbol esters, which activates protein kinase C (PKC), increased LRRK1-mediated Rab7A phosphorylation, and that this was inhibited by the treatment with Gö6983, a PKC inhibitor. Collectively, these reports imply that LRRK1 may be involved in the EGFR signaling pathway downstream of PKC via its kinase activity (Figure 4A).

Another study employing a system-wide spatiotemporal characterization of the ErbB-family receptor signaling complex identified LRRK1 in the EGFR receptor signaling complex as well as the cytosolic receptor-free subcomplex [82].

### 3.2. Rab7A Phosphorylation by TBK1

TBK1 (TANK-binding kinase 1) had been known to facilitate efficient mitophagy by phosphorylating optineurin (OPTN), Nuclear domain 10 protein 52 (NDP52)/Calcoco2 and Sequestosome-1 (SQSTM1), which are adaptors that link the ubiquitin chain formed on the mitochondrial outer membrane to the autophagy machinery. The TBK1-dependent phosphorylation of OPTN was shown to promote its binding to ubiquitin chains, while facilitating TBK1 activation, the retention of OPTN on mitochondria, and thereby efficient mitophagy, working as a positive feedback mechanism [83]. The impaired function and interaction of these proteins by genetic mutations has been linked with familial forms of amyotrophic lateral sclerosis (ALS) and fronto-temporal dementia (FTD) [84,85]. Of note, TBK1 is also known to phosphorylate LRRK2 in macrophages upon stimulation with Toll-like receptor ligands [86].

In 2018, Heo and colleagues further showed that upon mitochondrial depolarization, TBK1 phosphorylates the Ser72 residue of Rab7A in a Parkin-PINK1 dependent manner [34]. This residue is the same site as that phosphorylated by LRRK1 and is also homologous to the residue of Rab8 and Rab10 that is phosphorylated by LRRK2. It was proposed that Rab7A, upon being phosphorylated by TBK1 on the mitochondrial outer membrane, weakens its association with GDIs and avoids being extracted from the membrane. Instead, it recruits a different effector, the FLCN-FNIP1 complex, to the mitochondrial outer membrane, and enhances efficient mitophagy (Figure 4B). Considering that the loss of TBK1 function causes familial ALS, it would be interesting to see if the phosphorylation of Rab7A is altered in sporadic as well as familial ALS patients.

### 3.3. The Possible Involvement of Rab7A Phosphorylation in Tumor Progression

The phosphorylation of Rab7A was shown to be antagonized by PTEN, a dual-specific phosphatase that can dephosphorylate both lipid and protein substrates [87]. It was reported that PTEN dephosphorylates Rab7A at Ser72 and Tyr183; this dephosphorylation is required for the interaction of Rab7A with its activator proteins such as GDIs and GEF. The activation of Rab7A by PTEN facilitated the maturation of EGFR-containing endosomes and their degradation via fusion with lysosomes. This was significant not only in implicating the role that Rab7 plays in tumor progression, but also in exploring the function of PTEN as a tumor suppressor. It had been widely proposed that PTEN functions as a tumor suppressor through its lipid phosphatase activity, but these results further suggested that the protein phosphatase activity of PTEN is also important for the tumor suppressor function, through controlling the duration of EGF signaling.

A paper published in 2020 by Ritter and colleagues also reported the involvement of Rab7A phosphorylation in tumor progression [88]. It had been shown that Rab7A plays a role in attenuating receptor signaling in tumors, via trafficking the receptor/signaling molecule-containing vesicles to lysosomes and promoting their degradation. These include EGFR (as mentioned above) and stimulator of interferon genes (STING). STING activation results in the production of cytokines that activate innate immune signaling. The authors showed that in PTEN-null triple-negative breast cancer, when the phosphomimetic S72E mutant of Rab7A is overexpressed, hence when Rab7A cannot be activated, the degradation of STING is suppressed, and the production of cytokines, such as CCL5, CXCL10 and IFNβ, is aberrantly enhanced. Therefore, the phosphorylation of Rab7A may play a role in the hyperproduction of cytokines in tumor progression.

Collectively, it has been well established that the biological activity of Rab7A is regulated through phosphorylation by LRRK1 and TBK1, although it remains elusive whether there is a crosstalk between these two signals.

## 4. Conclusions

In recent years, new aspects of the regulation of the Rab function by phosphorylation have become apparent, some of which suggest a role in diseases such as PD, ALS and cancer. This review focuses on recently published articles, with a particular focus on phosphorylation in the switch II region of Rab proteins. It is desirable for future studies to shed light on the more detailed role of Rab phosphorylation in regulating its function, and on the crosstalk between disease-related signal transduction and intracellular vesicle trafficking, in which Rab kinases may play an important role.

## Figures and Tables

**Figure 1 biomolecules-11-01340-f001:**
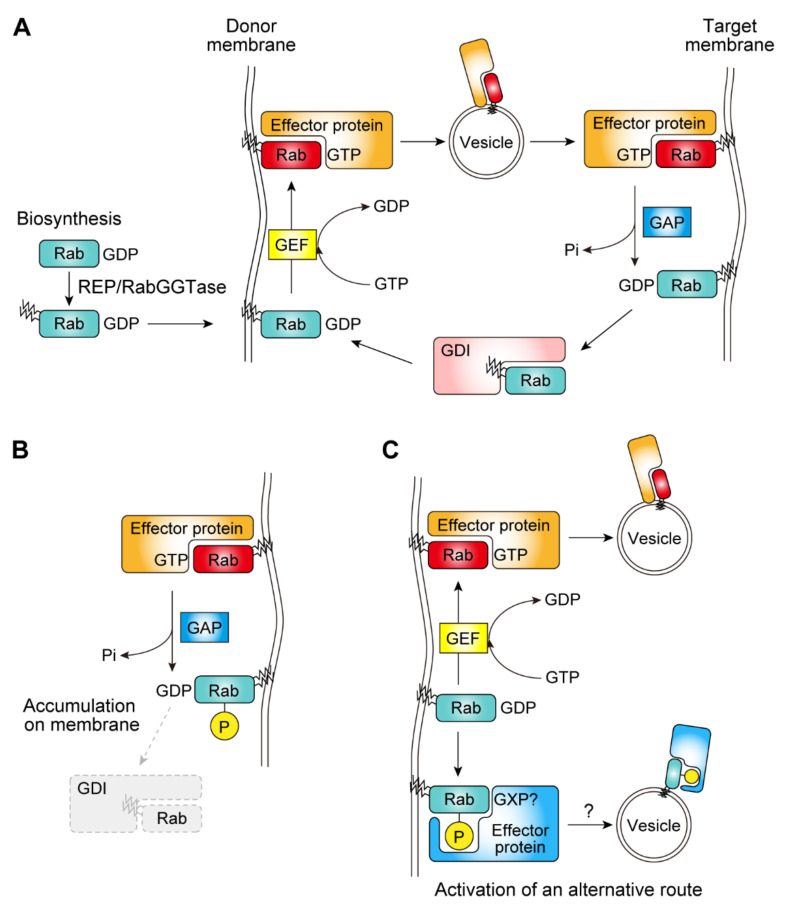
Schematic illustrations of the lifecycle of Rab proteins and proposed effects of phosphorylation. (**A**) The general life cycle of Rab proteins. (**B**) A potentially inhibitory effect of Rab phosphorylation on the extraction by GDP-dissociation inhibitors (GDIs). (**C**) An alternative route activated by Rab phosphorylation. REP: Rab escort protein; RabGGTase: Rab geranylgeranyl transferase; GEF: guanine nucleotide exchange factor; GAP: GTPase-activating protein; Pi: inorganic phosphate; GXP: GTP or GDP. ‘P’ in a yellow circle and blank yellow circles mean phosphorylation.

**Figure 2 biomolecules-11-01340-f002:**
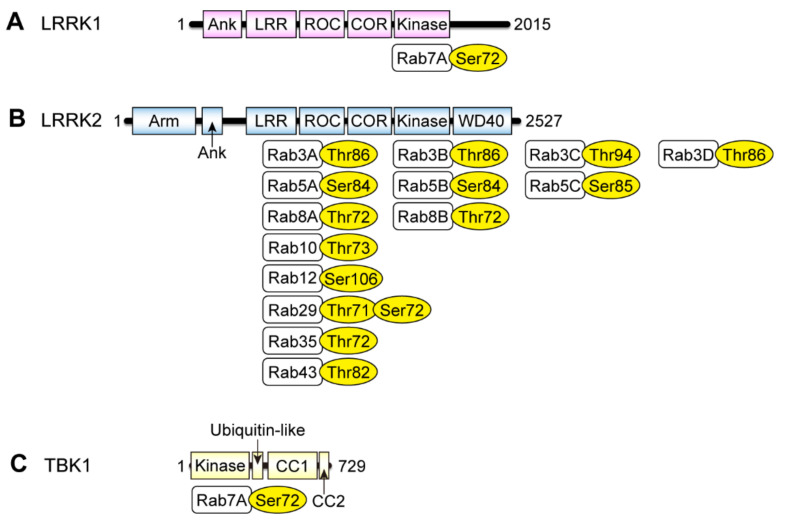
Domain architectures of the Rab kinases and their substrate Rab proteins. Domain architectures of (**A**) leucine-rich repeat kinase 1 (LRRK1), (**B**) leucine-rich repeat kinase 2 (LRRK2) and (**C**) TANK-binding kinase 1 (TBK1), and their substrate Rab proteins as well as the phosphorylation sites. Ank: ankyrin repeats; Arm: armadillo repeats; LRR: leucine-rich repeats; ROC: Ras of complex proteins; COR: carboxyl-terminal of ROC; WD40: WD40 repeats; CC: coiled-coil.

**Figure 3 biomolecules-11-01340-f003:**
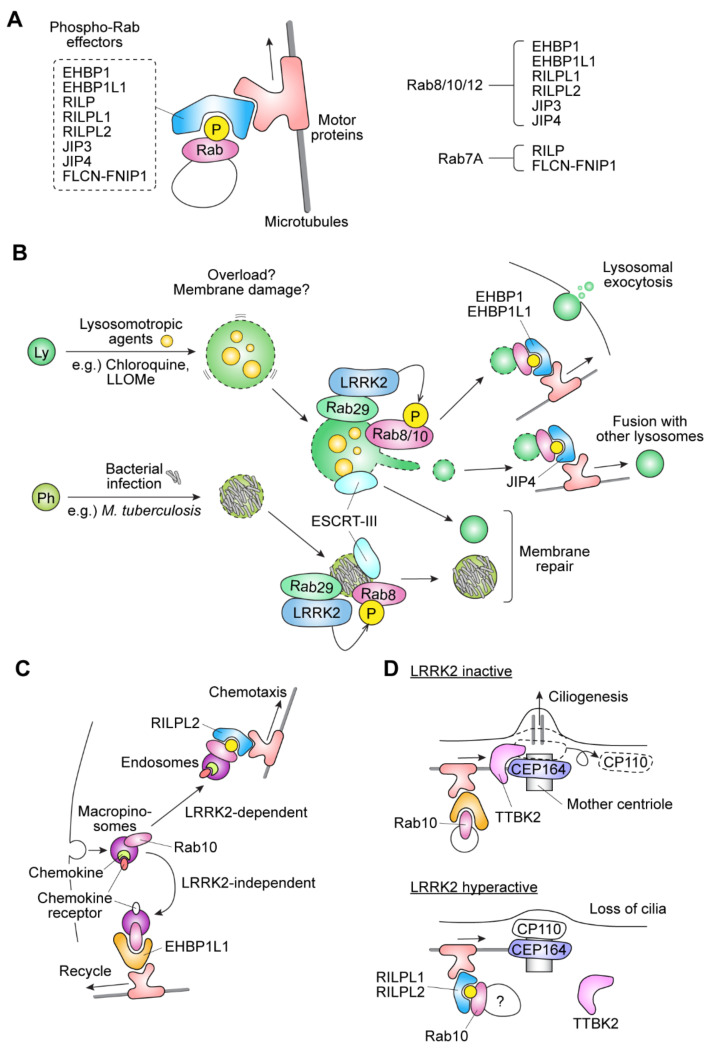
Proposed biological roles of Rab phosphorylation by LRRK2. (**A**) Proposed effectors of phosphorylated Rab proteins. They form complexes with motor proteins, thereby, in most cases except Folliculin (FLCN)-FLCN-interacting protein 1 (FNIP1), accelerating cargo transport. ‘P’ in a yellow circle and blank yellow circles mean phosphorylation. (**B**) Phosphorylation of Rab8/10 by LRRK2 plays important roles in macrophages under conditions such as lysosomal stress and bacterial infection. Ly: lysosomes; Ph: phagosomes. (**C**) Phosphorylation of Rab10 by LRRK2 functions as a molecular switch upon macropinocytosis; the macropinosomes are recycled back to the plasma membrane in a LRRK2-independent manner, whereas LRRK2-mediated phosphorylation of Rab10 activates a pathway in which the macropinosomes become maturated to signaling endosomes that somehow activate chemotactic pathways. (**D**) Aberrant phosphorylation of Rab10 by LRRK2 abnormally accelerates the transport of the Rab-interacting lysosomal protein-like protein 1/2 (RILPL1/2) cargo, the accumulation of which, in the pericentriolar region, might prevent centrosomal protein 164 (CEP164) and tau-tubulin kinase 2 (TTBK2) from localizing to the mother centriole, maintain centrosomal protein of 110 kDa (CP110) on the mother centriole and inhibit membrane transport necessary for ciliogenesis.

**Figure 4 biomolecules-11-01340-f004:**
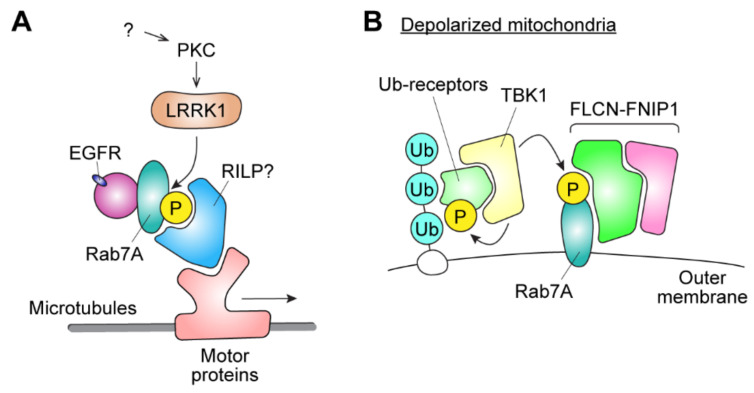
Proposed biological roles of Rab7A phosphorylation. (**A**) Phosphorylation of Rab7A by LRRK1 increases its affinity to RILP or other effectors, thereby facilitating the transport of EGFR-containing vesicles. Phosphorylation of Rab7A by LRRK1 is activated by PKC. (**B**) On the outer membrane of depolarized mitochondria, ubiquitylation of outer membrane proteins by Parkin recruits ubiquitin receptors such as optineurin. TBK1, which forms a complex with the Ub receptors, is recruited to the outer membrane and phosphorylates Rab7A as well as the receptors. Phosphorylated Rab7A recruits FLCN-FNIP1, thereby further promoting the mitophagy process.

## Data Availability

No new data were created or analyzed in this study.

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
