# Peer review of "The Regulation of Rab GTPases by Phosphorylation"

_biomolecules, 2021, doi:10.3390/biom11091340_

Round 1

Reviewer 1 Report

The review by Xu et al., is well written and informative. The topic is relevant. The figures are very good. I have some points below.

The abstract is missing a full stop toward the end.

In the section of the discovery of Rab phosphorylation by LRRK2, it is stated that a Lewy body appears in the remaining dopaminergic neurons. As worded this seems not correct. Many Lewy bodies are found throughout PD brain and Lewy pathology precedes dopaminergic neuron loss.

Italics are usually used when referring to genes.

The section on Rab phosphorylation by LRRK2 in human samples should include the studies of Atashrazm et al., Mov Disord 2019 and Wang et al., Sci Rep 2021. Are there any studies looking at LRRK2 mediated Rab phosphorylation in human tissues besides blood cells (ie brain).

In this sentence in the section of the proposed role of LRRK2 in a-syn propagation – “Moreover, none of the three LRRK2 inhibitors tested failed to 281 reduce α-synuclein pathology induced by the treatment of neurons with pre-formed α- 282 synuclein fibrils.” I think this meant to say all of the 3 inhibitors tested failed to reduce a-syn pathology? This section could also benefit from the authors providing more insight into why they think there are discrepancies. There are other studies suggesting LRRK2 inhibitors can reduce a-syn pathology that have not been cited, and importantly many of these are in human iPS cell models rather than mouse models.

In the section on Rab phosphorylation by TBK1, the authors could mention that TBK1 is also genetically linked to neurodegenerative diseases. TBK1 also phosphorylates LRRK2 at Ser935!

It is mentioned that there are many (60) Rab proteins but are these all ubiquitously expressed, or may certain Rabs have functions in certain cells?

LRRK2 and LRRK1 are both RIP kinase family members, could other Rip kinase family members (which are all stress-sensing kinases) also phosphorylate Rabs?

What is known about the dephosphorylation of Rab proteins? Phosphorylation is normally a reversible process but this does not seem to be covered in the review.

Reviewer 2 Report

Xu et al. have written a comprehensive review of the role that phosphorylation plays in regulating the activity of Rab GTPases and the implications for disease, Parkinson’s in particular. The recent discovery that LRRK2 phosphorylates a number of Rab GTPases has sparked increasing interest in what has traditionally been an understudied aspect of Rab biology. This is a very well written, timely and important review and I recommend that it be accepted for publication with the following very minor revisions.

Line 21: missing full stop.

Line 28: ‘The Rab family is a member…..’

Line 31: ‘Like other small GTPases, Rab GTPases exist….’

Lines 102/104: Type fourteen instead of 14.
